# Evolution Game Analysis of Hospital Governance Strategy in Industrial Parks in China

**DOI:** 10.3390/ijerph20043156

**Published:** 2023-02-10

**Authors:** Jie Zhen, Senmiao Yang

**Affiliations:** 1Business School, East China University of Political Science and Law, Shanghai 200042, China; 2Academy of Plateau Science and Sustainability, Qinghai Normal University, Xining 810016, China; 3School of Economics and Management, Tongji University, Shanghai 200092, China

**Keywords:** function stripping, hospital governance, industrial parks, evolutionary game

## Abstract

Industrial parks are an essential component of China’s reformation and opening, and they are the focus of sustainable economic and social development. However, in the process of further high-quality development, the relevant authorities have taken different approaches on whether to divest the social management functions of the parks, which introduces a dilemma of choice in reforming the management functions of these parks. This paper takes a comprehensive list of the hospitals providing public services in industrial parks as the representative subjects to clarify the factors influencing the selection of social management functions in industrial parks and the process in which they perform their roles. We also construct a tripartite evolutionary game model of the government, the industrial parks, and the hospitals and discuss the management functions of reform in industrial parks. The results show the following: (1) the selection of social management functions in industrial parks is an evolutionary game process of the government, park, and hospital under bounded rationality; (2) whether the government divests the park’s administrative authority over the hospital is affected by the cost of the government running the hospital and the additional benefits of the hospital’s participation in the business environment co-creation; (3) whether the industrial park provides high subsidies to the hospital is affected by its reputation benefit and subsidy cost; (4) whether the hospital participates in business environment co-creation is affected by additional benefits, subsidies, and its participation cost. When considering whether the local government should strip the social management function of the park over the hospital, it is not possible to simply “choose one of the two” or adopt a “one-size-fits-all” approach. Instead, attention should be paid to the factors influencing the choice of the main behaviors of all parties, the allocation of resources from the overall perspective of regional economic and social development, and jointly improving the business environment to achieve win–win results among all parties.

## 1. Introduction

The 20th National Congress of the Communist Party of China (CPC) emphasizes the realization of Chinese-style modernization, and the reformation and opening process of China over the past 40 years shows that industrial parks are the main organizational form with Chinese characteristics and reflect the common attributes of modernization of many other countries [1]. By the end of 2021, the gross domestic product (GDP) created by only 230 state-level economic and technological development zones and 169 state-level high-tech industrial development zones in China exceeded 1/5 of the national GDP in the same period. The total tax revenue was close to 1/4 of the national tax revenue in the same period [2]. There is no denying that industrial parks remain vital to the country’s efforts to optimize the business environment and achieve high-quality development in promoting Chinese-style modernization. Due to the development of industrial parks requiring decisive government intervention [3,4] and taking policies as an essential means [5], the local governments have carried out the reform of the system and mechanism of industrial parks one after another [6], significantly adjusting the management functions of industrial parks to meet the requirements of sustainable development.

The management function of industrial parks mainly includes two aspects: economic management function and social management function. The function of economic management primarily obtains economic performance, such as tax revenue, by encouraging enterprises in industrial parks. The role of social management is mainly to improve the business environment through supporting public services such as hospitals [2]. However, in the early stage of development, namely the early 1980s, China’s industrial parks mainly relied on economic management to promote industrial development and technological progress while ignoring the provision of public services and living space [7]. In this stage, social management functions such as medical and health care were weak [8] and were mainly undertaken by the local government [9]. In the period of stable reorganization after 2003 [2], the state adjusted the guiding ideology of the scientific development of industrial parks. It proposed to “promote the transformation of the industrial parks into multifunctional and comprehensive industrial areas,” or city functional areas combining industrial development and urbanization [10]. Then, it gradually paid attention to building social ecology in the parks. Especially in the last decade, the parks have positioned themselves as new urban areas with relatively comprehensive functions that mainly develop industry [11] and increasingly strengthened the construction of public facilities, living services, and industry–city integration [7,12,13].

This development trend indicates that the overall urbanization level of the region where the industrial park is located has been dramatically improved in a short time [14], and its urban characteristics continue to increase. Administrative authority and functions should be appropriately strengthened [15] to enhance park activities’ efficiency and establish a competitive operating mechanism [16]. In particular, the 18th National Congress of CPC put forward the value goal of a “better life”, which makes industrial parks pay more attention to the critical role of comprehensive hospitals in improving the business environment and attracting high-quality talents. Some scholars believe that this is the implementation of the change from industrial parks to administrative districts [11], indicating that park management is returning to the traditional system [17]. Some scholars regard the return as the reform and innovation of the conventional government management system [18]. Still, some other scholars think it is the loss of the advantages of the industrial park management system [19]. Specifically, the “government-oriented” management system weakens the park’s institutional innovation and efficient development functions. As a result, it is difficult for the park to continue to pioneer industrial upgrading and growth mode transformation [20], and it even brings problems such as increased organizations, personnel expansion, and departmental friction [8]. So, the industrial parks tend to concentrate on the economic management function, while the social management function has been stripped off and entrusted to the local government. On the one hand, the stripping or weakening of functions such as social management and public services, the transferring of social functions such as medical and health care to local governments or specific social organizations, and the highlighting of the economic management functions of the parks [21] are conducive to the industrial parks focusing on the primary responsibility of economic activities to a certain extent [22]. On the other hand, after the hospitals and other organizations providing social services are transferred to local governments, their supportive role in creating a business environment may be reduced, which is detrimental to the high-quality development of industrial parks and local economic and social development. It can be seen that the existing literature has begun to pay attention to the management function of industrial parks, but the documents have different views on the government’s approach to the reform of the management function and authority of industrial parks. In practice, the tasks and responsibilities of industrial parks have always been asymmetrical, ununified, and unstable [23], and there are contradictions between the economic and social management functions of industrial parks [8]. Whether the governments implement “re-administration” reform to strengthen social management functions or the “de-administration” reform to strip away the function of social management, related problems also arise.

The reform of the management function of industrial parks is not a process of “choosing one of the two”. Achieving unified reform results through the “one-size-fits-all” approach is challenging. Instead, it needs to adjust the authority–responsibility relationship between the industrial parks and administrative districts [24,25] as well as optimize the internal subjects and their interactive relationship with urban agglomeration [26] to improve the governance level to balance the economic management functions and social management functions of industrial parks [8] and clarify the scope of management functions of industrial parks [23]. Some scholars have recognized that such a balance depends on the innovation environment formed by multi-agents [27], requires cross-border resource integration [28], and brings about the overall dynamic effect of aggregation [29], which emphasizes the collaborative development of multiple subjects [30]. Specifically, industrial parks are a network of value creation and co-evolution formed by multiple subjects [31]. Through the policy platform created by the government, parks, and other entities [32], regional competitiveness can be enhanced [33,34]. These views represent the development orientation of the socialization of public service. Of course, from the legalization perspective, the legitimacy of social management functions such as medical and health services can also be clarified by redesigning the management system of the parks. However, this is a long-term and intractable problem [8].

On the whole, the existing literature suggests that the management system combining government and district should be adopted [35], or the management system combining “administrative district and functional district” should be implemented [9,19]. At the same time, it is also believed that various types of industrial parks have their advantages and disadvantages and should not blindly “follow the trend” to change to a certain kind of management system but should adopt the most suitable management system for its current development stage according to its actual situation, to clarify its functional management scope [35]. Moreover, it is challenging to complete the transformation of the parks’ management functions only through reforming the administrative system [10]. In terms of the supply of public goods, it is the internal demand of the transformation of the system to assume both economic management functions and social management functions [36], so it is necessary to pay attention to the diversified supply and cooperative governance of the government, market, and society [37]. These research results further point out the necessity of in-depth analysis of the actual development of each main subject.

However, although the current researchers adopt the Malmquist index [35], system dynamics [38], qualitative comparative analysis of fuzzy sets [39], and other research methods to analyze the types and functions of management systems, they have relatively neglected the characteristics of bounded rationality of each subject in the process of industrial parks development. In fact, the reform of the parks management functions involves the game relationship among various subjects such as local governments, industrial parks, and hospitals [9]. Moreover, the subjects are not entirely rational, and there are win–win interests and conflicts of interest among them when they make strategic choices. How to make the best decision in line with their interests in this complex environment is a problem that local governments, industrial parks, and hospitals must consider. As a game theory based on the bounded rationality of game players, the evolutionary game combines game theory analysis with the study of the dynamic evolution processes, emphasizing a dynamic equilibrium. Because evolutionary game makes up for many defects of traditional game theory, it is widely used in various fields such as economic management and operation. Presently, hospital governance strategies in China mainly focus on the high-quality development of comprehensive public hospitals, and the construction of regional medical centers is an important evaluation index and direction for hospitals’ innovative development [40]. While the comprehensive public in the industrial parks will become the medical center of the parks due to their unique geographical location, serving enterprise employees and other patients. It can be seen that how to improve the quality of hospital services and carry out innovative development is a problem that the hospital, the park, and the government need to think about. Therefore, this paper adopts the evolutionary game method and fully considers the bounded rationality and strategy choice of each subject [41], then constructs a three-party evolutionary game model of the government, the industrial park, and the hospital to analyze the mutual feedback of all parties and the evolutionary stability strategy in the process of the government system and mechanism reform. This study aims to fully consider the role of the hospital and other social service providers in the co-creation of the business environment, pay attention to the ownership of administrative authority of the hospital and the influence of the strategic choices of each subject on the business environment, then solve the corresponding evolutionary stability points of the three parties by analyzing the benefits and costs of each party under different strategies to provide beneficial policy suggestions for reforming the management functions of industrial parks and their high-quality development.

## 2. Problem Description and Model Building

### 2.1. Problem Description

The theory of the evolutionary game is based on bounded rationality. As decision-makers are limited by time and ability, it is difficult to obtain complete long-term decision information, so they can only adjust their strategies according to the current income information [24]. Hence, this theory focuses on the mutual feedback of driving forces between different decision-makers and specifies the stable strategies of the evolution [25], which has been widely used in inter-agent decision selection and dynamic evolution. With the advance of the reform of the industrial park system and mechanism, governments in different regions of China have taken different decisions on determining the scope of park management functions within their jurisdiction. Considering that industrial parks should return to the primary responsibility and business, the local government stripped the social management function of the parks so that the parks can focus more attention and resources on economic development. However, in the context of the 19th National Congress of CPC, which called for “constantly meeting the people’s growing needs for a better life”, people are attaching greater importance to health while holding that the management authority of the hospital in the industrial park can bring convenience and high-quality medical services to the managers of various enterprises in the park, thus contributing to the co-creation of the business environment. Therefore, this paper constructs an evolutionary game model to explore the interaction among local governments’ willingness to adjust the administrative authority of hospitals in industrial parks, industrial parks’ willingness to subsidize hospitals’ participation in business environment co-creation, and hospitals’ desire to participate in business environment co-creation under the background of business environment co-creation. Moreover, this paper analyzes the influence of several factors, including the system mechanism reform and incentives to various subjects.

To sum up, the logical relationship of the triple game players in the business environment co-creation system under the background of park management function reform is shown in Figure 1, where x represents the probability that the local government will not divest the hospital from the industrial park.

### 2.2. Model Assumption

In the research on the strategic choice of participants in the business environment co-creation system, we mainly focus on the strategic choices of three stakeholders: the government, the industrial park, and the hospital. Therefore, when building an evolutionary game model, it is necessary to fully consider the strategic choices and benefits of the participants to study how to improve cooperation among stakeholders and achieve a win–win situation for all parties.

The government, the industrial park, and the hospital are considered bounded rationality. The government influences the business environment mainly by reforming the management authority of the industrial park, that is, whether to divest the management authority of the park to the local government. The government can choose either no divesture or divestiture strategies $\left(\mathrm{ND},D\right)$ in the strategy sets. The industrial park will consider whether to implement high subsidies to the hospital based on the benefits brought by the hospital’s sustainable development and business environment optimization, with its strategy choices of high subsidies and low subsidies $\left(\mathrm{HS},\mathrm{LS}\right)$. The hospital decides whether to participate in the co-creation of a business environment based on its development and the measures of the government and the industrial park so that it can choose either participation or nonparticipation strategies $\left(P,\;\mathrm{NP}\right)$ in the strategic sets.*Probability.* In the model, three decision-makers choose their strategies according to their desires. If the proportion of the government’s “ND” is denoted by x, then the proportion of the government’s “D” is 1−x. Moreover, if using f represents the proportion of choosing “HS” for the industrial park, then the proportion of “LS” for the industrial park is 1−f. Similarly, z and 1−z are the proportions of “P” and “NP” for the hospital, respectively, ($x,\;f,\;z\;\in \;\left[0,\;1\right]$).*The government.* When the government chooses strategy “ND”, the hospital is mainly managed by the park, so the government’s management cost $C_{gl}$ is low. If “D” is the government’s strategy, the management cost $C_{gh}$ is high. What is more, to increase support for business environment co-creation, the government will provide additional subsidies $M_{h}$ to the hospital if it chooses to participate in building a business environment.*The industrial park.* The cost of the “HS” choice of the industrial park is $B_{ph}$ and the cost of the “LS” choice is $B_{pl}$. When the government chooses the “ND” strategy, the hospital’s participation in the business environment co-creation will strengthen its interaction and cooperation with the park, so the park’s management cost $C_{pl}$ for the hospital is low. If the hospital chooses the “NP” strategy, the interaction between the hospital and the park will be weakened, and the management cost $C_{ph}$ of the park for the hospital is high. Meanwhile, we define S as the industrial park’s reputation gains if it chooses “HS” in the condition that the hospital chooses “P” strategy because when the park provides high subsidies to hospitals, the hospital participates in the co-creation of the business environment can provide specific high-level medical and health services under the guidance of the park, so as to improve the investment management effectiveness of the park and bring reputation benefits to the park. When the government’s strategy is “D” and the hospital chooses “NP”, there are no costs and benefits associated with the industrial park.*The hospital.* In general, the hospital’s usual revenue is $R_{h}$. When participating in the business environment co-creation, the hospital can provide special services (physical examination, medical convenience, etc.) for the employees of enterprises in the industrial park and will bring additional benefits to itself and its management body. That is, $R_{p}$ is the revenue of the park and the hospital when the government’s choice is “ND”, while when the government chooses “D”, $R_{p}$ is the revenue of the government departments and the hospital. When participating in business environment co-creation, hospitals will invest capital and management costs to meet the requirements of optimizing the business environment, resulting in additional costs $C_{h}$.

Related model parameters and their settings are shown in Table 1.

Based on the above assumptions, the payoff matrix for these stakeholders of the business environment co-creation is shown in Table 2.

## 3. Analysis

### 3.1. Evolutionary Stable Strategy of the Government

Let $U_{G1}$ and $U_{G2}$ represent expected income with the no divesture (“ND”) and strategy divesture (“D”), respectively, with the average expected income denoted by $\bar{U_{G}}$, which can be expressed by Equation (1).
(1)$$\left\{\begin{array}{c} U_{G1}=-zM_{h}-C_{gl}\;\\ U_{G2}=zR_{p}-C_{gh}-zM_{h}\\ \bar{\;U_{G}}=xU_{G1}+\left(1-x\right)U_{G2} \end{array}\right.$$

Therefore, the dynamic evolution equation of the government can be formulated as follows:(2)$$F\left(x\right)=\frac{dx}{dt}=x\left(U_{G1}-\bar{U_{G}}\right)=x\left(1-x\right)(-zR_{p}+C_{gh}-C_{gl})$$
(3)$$F^{\prime }\left(x\right)=\left(1-2x\right)(-zR_{p}+C_{gh}-C_{gl})$$

Based on Equations (1)–(3), if $F\left(x\right)=0$ and $F^{\prime }\left(x\right)<0$, the government that chooses the no divesture (“ND”) strategy would be in a stable state.

**Proposition** **1.**
*If $z<z^{*}$, where the threshold $z^{*}=\frac{-C_{gh}+C_{gl}}{-R_{p}}$, no divesture (“ND”) is a stable strategy for the government; conversely, if $z>z^{*}$, the government tends to choose the divesture (“D”) strategy; and if $z=z^{*}$, the stable strategy is unable to be determined. Proposition 1 implies that the industrial park’s authority to manage the hospital that participates in the business environment co-creation will be more likely to be divested by the government to increase its benefits. As shown in Figure 2, the volumes $V_{A1}$ and $V_{A2}$, respectively, represent the probability of the government choosing the “ND” strategy and the “D” strategy.*



(4)
$$V_{A1}=\int_{0}^{1}\int_{0}^{1}\frac{-C_{gh}+C_{gl}}{-R_{p}}dxdf=\frac{C_{gh}-C_{gl}}{R_{p}}$$



(5)
$$V_{A2}=1-V_{A1}=1-\frac{C_{gh}-C_{gl}}{R_{p}}$$


**Corollary** **1.**
*$\frac{\partial V_{A1}}{\partial C_{gh}}>0$, $\frac{\partial V_{A1}}{\partial C_{gl}}<0$, $\frac{\partial V_{A1}}{\partial R_{p}}<0$. The probability that the government does not divest the management authority of the park is positively correlated with the high hospital management cost $C_{gh}$ but negatively correlated with the low hospital management cost $C_{gl}$ and the additional benefits $R_{p}$.*


Corollary 1 indicates that when the government considers whether to divest the industrial park from managing the hospital, if the government’s management cost of the hospital after the divestment is high, the government will prefer not to divest, that is, to let the park manage the hospital to reduce its cost. In this case, the probability that the government will not divest the park’s management authority will increase. However, if the hospital is managed by the park and the management cost $C_{gl}$ invested by the government increases, or if the additional benefits $R_{p}$ of the hospital and its management body increases when the hospital participates in the co-creation of the business environment, the probability of the government choosing to divest the park’s management authority will increase.

### 3.2. Evolutionary Stable Strategy of the Industrial Park

Let the expected payoff of the industrial park to choose the high subsidies (“HS”) strategy be $U_{P1}$, the expected payoff of the low subsidies (“LS”) strategy be $U_{P2}$, and the average expected income be denoted by $\bar{U_{P}}$, which can be expressed by Equation (6).
(6)$$\begin{array}{c} U_{P1}=zxR_{p}+zS-xzC_{pl}-x\left(1-z\right)C_{ph}-\left[z+\left(1-z\right)x\right]B_{ph}\;\\ U_{P2}=zxR_{p}-xzC_{pl}-x\left(1-z\right)C_{ph}-\left[z+\left(1-z\right)x\right]B_{pl}\;\\ \bar{U_{P}}=fU_{P1}+\left(1-f\right)U_{P2}\; \end{array}$$

Therefore, the dynamic evolution equation of the industrial park can be formulated as:(7)$$F\left(f\right)=\frac{df}{dt}=f\left(U_{P1}-\bar{U_{P}}\right)=f\left(1-f\right)\left\{zS-\left[z+\left(1-z\right)x\right](B_{ph}-B_{pl}\right\}$$
(8)$$F^{\prime }\left(f\right)=\left(1-2f\right)\left\{zS-\left[z+\left(1-z\right)x\right](B_{ph}-B_{pl})\right\}$$

Based on Equations (6)–(8), if $F\left(f\right)=0$ and $F^{\prime }\left(f\right)<0$, the industrial park that chooses the high subsidies (“HS”) strategy would be in a stable state.

**Proposition** **2.**
*If $x>x^{*}$, where the threshold $x^{*}=\frac{z(S-B_{ph}+B_{pl})}{\left(1-z\right)\left(B_{ph}-B_{pl}\right)}$, high subsidies (“HS”) is a stable strategy for the industrial park, while the stable strategy would be the low subsidies (“LS”) if $x<x^{*}$; and if $x=x^{*}$, it is unable to be determined.*


Proposition 2 implies that if the government has a strong willingness to choose the no divesture strategy, the industrial park will provide high subsidies to the hospital because it can obtain additional benefits if the hospital decides to participate in the co-creation of the business environment, prompted by the industrial park’s high subsidies. Otherwise, when the government chooses the divesture strategy, the industrial park that is not able to achieve additional benefits prefers to offer low subsidies to the hospital in order to reduce its cost. In addition, Figure 3 shows the phase diagram of the strategic choice of the industrial park, where the volume $V_{B1}$, $V_{B2}$ represent the probability of the “LS” strategy and the “HS” strategy, respectively.
(9)$$V_{B2}=\int_{0}^{1}\int_{0}^{\frac{B_{ph}-B_{pl}}{S}}\frac{z(S-B_{ph}+B_{pl})}{\left(1-z\right)\left(B_{ph}-B_{pl}\right)}dzdf=-\frac{S-(B_{ph}-B_{pl})}{B_{ph}-B_{pl}}\left(\frac{B_{ph}-B_{pl}}{S}+\ln\frac{S-(B_{ph}-B_{pl})}{S}\right)$$
(10)$$V_{B1}=1-V_{B2}=1+\frac{S-(B_{ph}-B_{pl})}{B_{ph}-B_{pl}}\left(\frac{B_{ph}-B_{pl}}{S}+\ln\frac{S-(B_{ph}-B_{pl})}{S}\right)$$

**Corollary** **2.**
*The probability that the industrial park gives high subsidies to the hospital is positively correlated with the reputation benefits S but negatively correlated with the industrial park’s subsidy costs $B_{ph}$ and $B_{pl}$, where $S>B_{ph}-B_{pl}$.*


**Proof.** Firstly, according to Equation (9), $\ln\frac{S-(B_{ph}-B_{pl})}{S}$ makes sense only when $\frac{S-(B_{ph}-B_{pl})}{S}>0$; therefore, $S>B_{ph}-B_{pl}$.Secondly, find the first partial derivatives of each element in $V_{B2}$.$\frac{\partial V_{B2}}{\partial S}=\frac{S-(B_{ph}-B_{pl})}{S^{2}}+\frac{1}{S-(B_{ph}-B_{pl})}-\frac{B_{ph}-B_{pl}}{S}-\ln\frac{S-(B_{ph}-B_{pl})}{S}\geq \frac{2}{S}-\frac{B_{ph}-B_{pl}}{S}-\ln\frac{S-(B_{ph}-B_{pl})}{S}\geq -\frac{B_{ph}-B_{pl}}{S}-\ln\frac{S-(B_{ph}-B_{pl})}{S}$. Let $t=\frac{B_{ph}-B_{pl}}{S}\in \left(0,1\right)$, then $-\frac{B_{ph}-B_{pl}}{S}-\ln\frac{S-(B_{ph}-B_{pl})}{S}=f\left(t\right)=-t-ln\left(1-t\right)$, because $\frac{df\left(t\right)}{dt}=-1+\frac{1}{1-t}>0$ and $f\left(0\right)=0$, so $f\left(t\right)>0$, $\frac{\partial V_{B2}}{\partial S}>0$;$\frac{\partial V_{B2}}{\partial B_{ph}}=\frac{\partial V_{B2}}{\partial B_{pl}}=\frac{S}{{\left(B_{ph}-B_{pl}\right)}^{2}}\ln\frac{S-(B_{ph}-B_{pl})}{S}+\frac{1}{S}-\frac{S}{\left(S-B_{ph}+B_{pl}\right)\left(B_{ph}-B_{pl}\right)}=\frac{S}{{\left(B_{ph}-B_{pl}\right)}^{2}}\ln\frac{S-(B_{ph}-B_{pl})}{S}-\left[\frac{S-(B_{ph}-B_{pl})}{S\left(B_{ph}-B_{pl}\right)}+\frac{1}{S-(B_{ph}-B_{pl})}\right]\leq \frac{S}{{\left(B_{ph}-B_{pl}\right)}^{2}}\ln\frac{S-(B_{ph}-B_{pl})}{S}-2\sqrt{\frac{1}{S\left(B_{ph}-B_{pl}\right)}}<0$. □

Corollary 2 reveals that when the industrial park gives high subsidies to the hospital participating in the co-creation of the business environment and thus obtains high reputation benefits, enterprises will be more recognized for the social service level of the industrial park, which is conducive to the park carrying out important work, such as investment attraction, and effectively promoting its high-quality development. Therefore, the park is more willing to offer high subsidies. However, with the increase in subsidy cost, the park’s willingness to give high subsidies gradually reduces.

### 3.3. Evolutionary Stable Strategy of the Hospital

Let the expected payoff of the hospital to choose the participation (“P”) strategy be $U_{H1}$, the expected payoff of the nonparticipation (“NP”) strategy be $U_{H2}$, and the average expected income be denoted by
$\bar{U_{H}}$, which can be expressed by Equation (11).
(11)$$\begin{array}{c} U_{H1}=R_{h}+R_{p}+M_{h}-C_{h}+fB_{ph}+\left(1-f\right)B_{pl}\\ U_{H2}=R_{h}+xfB_{ph}+x\left(1-f\right)B_{pl}\;\\ \bar{U_{H}}=zU_{H1}+\left(1-z\right)U_{H2}\; \end{array}$$

Therefore, the dynamic evolution equation of the hospital can be formulated as follows:(12)$$F\left(z\right)=\frac{dz}{dt}=z\left(1-z\right)\left\{R_{p}+M_{h}-C_{h}+\left(1-x\right)\left[fB_{ph}+\left(1-f\right)B_{pl}\right]\right\}$$
(13)$$F^{\prime }\left(z\right)=\left(1-2z\right)\left\{R_{p}+M_{h}-C_{h}+\left(1-x\right)\left[fB_{ph}+\left(1-f\right)B_{pl}\right]\right\}$$

Based on Equations (11)–(13), if $F\left(z\right)=0$ and $F^{\prime }\left(z\right)<0$, the hospital that chooses the participation (“P”) strategy would be in a stable state.

**Proposition** **3.**
*If $x<x^{*}$, where the threshold $x^{*}=1+\frac{R_{p}+M_{h}-C_{h}}{fB_{ph}+\left(1-f\right)B_{pl}}$, participation (“P”) is a stable strategy for the hospital; conversely, if $x>x^{*}$, the hospital tends to choose the nonparticipation (“NP”) strategy; and if $x=x^{*}$, the stable strategy is unable to be determined. In Figure 4, The volume $V_{C1}$ and $V_{C2}$, respectively, represent the probability of the hospital choosing participation and nonparticipation strategies.*



(14)
$$V_{C1}=\int_{0}^{1}\int_{0}^{1}1+\frac{R_{p}+M_{h}-C_{h}}{fB_{ph}+\left(1-f\right)B_{pl}}dfdz=1+\frac{R_{p}+M_{h}-C_{h}}{B_{ph}-B_{pl}}\ln\frac{B_{ph}}{B_{pl}}$$



(15)
$$V_{C2}=1-V_{C1}=-\frac{R_{p}+M_{h}-C_{h}}{B_{ph}-B_{pl}}\ln\frac{B_{ph}}{B_{pl}}$$


**Corollary** **3.**
*The probability of a hospital’s stable participation in business environment co-creation is positively related to the additional benefits $R_{p}$ and the government subsidy $M_{h}$, negatively correlated with the hospital’s participation cost $C_{h}$, and affected by industrial park subsidies $B_{ph}$ and $B_{pl}$.*


**Proof.** According to Equation (14), calculate the first partial derivatives of each element in $V_{C1}$:


$$\frac{\partial V_{C1}}{\partial R_{p}}=\frac{\partial V_{C1}}{\partial M_{h}}=\frac{1}{B_{ph}-B_{pl}}\ln\frac{B_{ph}}{B_{pl}}>0\;\frac{\partial V_{C1}}{\partial C_{h}}=-\frac{1}{B_{ph}-B_{pl}}\ln\frac{B_{ph}}{B_{pl}}<0\;\frac{\partial V_{C1}}{\partial B_{ph}}=\frac{R_{p}+M_{h}-C_{h}}{B_{ph}-B_{pl}}\left(-\frac{1}{B_{ph}-B_{pl}}\ln\frac{B_{ph}}{B_{pl}}+\frac{1}{B_{ph}}\right)=\frac{R_{p}+M_{h}-C_{h}}{{\left(B_{ph}-B_{pl}\right)}^{2}}\left(1-\ln\frac{B_{ph}}{B_{pl}}-\frac{B_{pl}}{B_{ph}}\right)=\frac{R_{p}+M_{h}-C_{h}}{{\left(B_{ph}-B_{pl}\right)}^{2}}\left(\ln\frac{eB_{pl}}{B_{ph}}-\ln e^{\frac{B_{pl}}{B_{ph}}}\right)\;\frac{\partial V_{C1}}{\partial B_{pl}}=\frac{R_{p}+M_{h}-C_{h}}{B_{ph}-B_{pl}}\left(\frac{1}{B_{ph}-B_{pl}}\ln\frac{B_{ph}}{B_{pl}}-\frac{1}{B_{pl}}\right)=\frac{R_{p}+M_{h}-C_{h}}{{\left(B_{ph}-B_{pl}\right)}^{2}}\left(1+\ln\frac{B_{ph}}{B_{pl}}-\frac{B_{ph}}{B_{pl}}\right)=\frac{R_{p}+M_{h}-C_{h}}{{\left(B_{ph}-B_{pl}\right)}^{2}}\left(\ln\frac{eB_{ph}}{B_{pl}}-\ln e^{\frac{B_{ph}}{B_{pl}}}\right)$$


There are three cases to be discussed in the above two equations: when $R_{p}+M_{h}-C_{h}>0$, $\frac{\partial V_{C1}}{\partial B_{ph}}<0$,$\frac{\partial V_{C1}}{\partial B_{pl}}<0$;when $R_{p}+M_{h}-C_{h}<0$, $\frac{\partial V_{C1}}{\partial B_{ph}}>0$,$\frac{\partial V_{C1}}{\partial B_{pl}}>0$;when $R_{p}+M_{h}-C_{h}=0$, $\frac{\partial V_{C1}}{\partial B_{ph}}=\frac{\partial V_{C1}}{\partial B_{pl}}=0$. □

Obviously, when the additional benefits or subsidies given by the government increase, the hospital tends to participate in business environment co-creation because it can obtain more benefits. When the cost of participation is high, the hospital is more likely to choose not to participate in order to reduce expenditure. In addition, when $R_{p}+M_{h}$ is greater than the hospital’s participation cost $C_{h}$, the probability of the hospital’s participation in the co-creation may decrease with the increase in the industrial park’s subsidy because, at this time, the park’s requirements for hospitals are also increased, and the hospital needs to meet the higher requirements put forward by the park. This increases the hospital’s pressure and reduces its willingness to participate in the co-creation of the business environment. Similarly, when $R_{p}+M_{h}$ is less than the hospital participation cost $C_{h}$, the willingness of the hospital to participate in the co-creation will be enhanced with the increase in the industrial park’s subsidy.

## 4. Evolutionary Strategic Portfolio Analysis and Simulation

### 4.1. Stability Analysis of Evolutionary Strategic Portfolios

The differential equation system describes the group dynamics, and the stability of the strategy of the game can be judged by the Lyapunov theory [26]. The Jacobian matrix is used to analyze the ESS of the differential equation system. First, by calculating the eight equilibrium points in this dynamic evolutionary game system, the below Jacobian matrix is obtained.
(16)$$J=\left[\begin{array}{ccc} \partial F\left(x\right)/\partial x & \partial F\left(x\right)/\partial f & \partial F\left(x\right)/\partial z\\ \partial F\left(f\right)/\partial x & \partial F\left(f\right)/\partial f & \partial F\left(f\right)/\partial z\\ \partial F\left(z\right)/\partial x & \partial F\left(z\right)/\partial f & \partial F\left(z\right)/\partial z \end{array}\right]$$

Then, we analyze the stability of the eight equilibrium points in the model. Mark the distinct positive eigenvalues as (+), the negative eigenvalues as (−), and the uncertain eigenvalues as (u), as shown in Table 3. According to the EGT, the equilibrium points that satisfy all eigenvalues of the Jacobian matrix being non-positive are the evolutionary stable points, while those points which contain at least one eigenvalue greater than zero are unstable.

According to Table 3, $\left(0,\;1,\;1\right)$, $\left(1,\;0,\;0\right)$, and $\left(1,\;1,\;1\right)$ are possible, stable strategies. Firstly, under Condition 1, $\left(0,\;1,\;1\right)$, namely divesture, high subsidies, and participation, will be the stable outcome of the system. This suggests that when the benefits the hospital can obtain by participating in the co-creation of the business environment is larger than the cost, the hospital will finally choose the participation strategy, and the industrial park is willing to offer high subsidies to accelerate the process. However, since the additional benefits $R_{p}$ are large enough to cover the difference between the government’s management cost when choosing different strategies, it can be expected that the government will choose to divest the industrial park’s management authority of the hospital to gain the additional benefits.

Secondly, $\left(1,\;0,\;0\right)$, namely no divesture, low subsidies, and nonparticipation, is the stable outcome under Condition 2. This indicates that if the cost of participating in the establishment of a business environment is too high, there is no incentive for the hospital to take such action, as well as for the industrial park to sponsor with high subsidies. Meanwhile, the government will naturally choose the no divesture strategy, which means it will let the industrial park take charge of the hospital to prevent high management costs.

At last, when Condition 2 is met, $\left(1,\;1,\;1\right)$ is going to be the stable result of the system; that is, the strategic portfolios are no divesture, high subsidies, and participation. This implies that under the condition of the hospital’s participation in establishing a business environment, only when the difference between the two strategies’ management cost is higher than possible benefits will the government eventually give up taking charge of the hospital.

### 4.2. The Probabilities of the Game Players’ Strategy Choices

For the purpose of further verifying how the three parties’ strategy choices are influenced, we use MATLAB 2022 to simulate the changes in the choice probability when the government, the industrial park, and the hospital choose no divesture (“ND”), high subsidies (“HS”), and participation (“P”) strategies, respectively.

#### 4.2.1. Choice Possibility When the Government Chooses the “ND” Strategy

The relationship between the probability of the government choosing the “ND” strategy and $R_{p}$, $C_{gh}$, $C_{gl}$ is shown in Figure 5. Based on Formula (4), let $\delta =C_{gh}-C_{gl}$ for the sake of illustration, then $V_{A1}=\frac{\delta }{R_{p}}$. Since $C_{gh}>C_{gl}$, we suppose $\delta \in \left[5,\;30\right]$, $R_{p}\in \left[20,\;40\right]$.

As can be seen from Figure 5, when the difference between the management cost of the government’s two strategies is high, or the additional revenue from the hospital’s participation in the co-creation of the business environment is low, the government cannot obtain many benefits from managing the hospital, so it intends to choose no divesture strategy. Otherwise, the divesture strategy is the government’s optimal choice.

#### 4.2.2. Choice Possibility When the Industrial Park Chooses the “HS” Strategy

The relationship between the probability of the industrial park choosing the “HS” strategy and S, $B_{ph}$, $B_{pl}$ is shown in Figure 6. Based on Formula (9), set $\Lambda =B_{ph}-B_{pl}$ for clear illustration, then $V_{B2}=-\frac{S-\Lambda }{\Lambda }\left(\frac{\Lambda }{S}+\ln\frac{S-\Lambda }{S}\right)$ Because S>Λ, we make $S\in \left[0,\;50\right]$, $\Lambda \in \left[0,\;20\right]$.

According to Figure 6, with the increase in the difference between the reputation benefits S and Λ, the probability of the industrial park choosing the “HS” strategy gradually increases. However, the further growth of the difference will decrease the possibility of the “HS” strategy chosen by the park, which may be because the continuous increase in reputation benefits is affected by other factors in addition to the subsidy.

#### 4.2.3. Choice Possibility When the Hospital Chooses the “P” Strategy

Shown in Figure 7 is the relationship between the probability of the hospital choosing the “P” strategy and $B_{ph}$, $B_{pl}$. Based on Formula (14), let $\Delta =R_{p}+M_{h}-C_{h}$ for clear illustration, then $V_{C1}=1+\frac{\Delta }{B_{ph}-B_{pl}}\ln\frac{B_{ph}}{B_{pl}}$. Since we cannot tell Δ is positive or negative, we need to make a categorical discussion by letting Δ=±5, $B_{ph}\in \left(0,15\right]$,$B_{pl}\in \left(0,10\right]$.

According to Figure 7, as long as Δ>0, namely, the hospital’s participation cost is less than the sum of the additional benefits and the government’s subsidy, the participation strategy of the hospital is dominant no matter what kind of subsidy the park gives to the hospital. If Δ<0, the probability of the hospital’s participation will increase with the rise in the park’s subsidy.

### 4.3. Numerical Simulation

We aim to study the evolution process of the strategic choice of participants in the co-creation of the business environment and the key factors affecting their behavior, predict future trends, and provide a theoretical basis for formulating relevant policies. In doing so, MATLAB 2022 software is adopted to simulate the evolutionary process dynamically, as shown in Figure 8, Figure 9, Figure 10, Figure 11, Figure 12 and Figure 13. According to daily practical business transaction examples, game models, and formulas, the experimental parameters are shown in Table 4. On this basis, the influence of additional benefits $R_{p}$, the hospital’s participation cost $C_{h}$, the government’s management costs $C_{gh}$ and $C_{gl}$, the government’s subsidy cost $M_{h}$, and the industrial park’s reputation benefits S on the process and result of the evolutionary game are analyzed.

A variable control approach is taken to analyze the impact of additional benefits $R_{p}$ by letting $R_{p}=\left\{5,\;30,\;60\right\}$, and the simulation results of the dynamic replication equations evolving 50 times over time are shown in Figure 8. With the increase in $R_{p}$, the final stable point of system evolution is different. When $R_{p}$ is small, the government will eventually choose not to divest the industrial park’s management authority of the hospital, so the industrial park has no incentive to invest a lot in the hospital, and the hospital will not participate in the co-creation of the business environment. With the increase in $R_{p}$, the industrial park is willing to give a high subsidy to the hospital to improve the income, and the hospital will also choose to participate in environmental co-creation. However, when $R_{p}$ continues to increase, the government will choose the divesture strategy, and it will be responsible for hospital management and share the additional income created by the hospital’s participation. Obviously, the size of the additional benefits has an essential impact on the strategic choices of the three parties in the game. Under the background of business environment optimization receiving more and more attention, the hospital should continuously increase the additional benefits by improving its governance. At the same time, the government should deal with the relationship with the industrial park from the overall perspective of regional economic and social development to look at the allocation of additional income and avoid unilateral benefit orientation.

When the hospital’s participation cost $C_{h}=\left\{10,\;40,\;60\right\}$, we can obtain the simulation results of strategy choice, as shown in Figure 9. When $C_{h}$ is small, the system evolves faster and finally stabilizes at $\left(x,f,z\right)=\left(0,\;1,\;1\right)$. Small $C_{h}$ means the hospital is willing to participate in the co-creation of the business environment, and the industrial park is willing to provide high subsidies to hospitals. However, in this case, the government may choose the divesture strategy to obtain additional benefits. On the contrary, when $C_{h}$ is large, the system evolution is stable at $\left(x,f,z\right)=\left(1,\;0,\;0\right)$; that is, because of the high cost, the hospital chooses not to participate, so the industrial park does not want to carry out high subsidies and the government is not willing to manage the hospital. Therefore, in order to encourage the hospital to participate in business environment co-creation, the government, especially the industrial park, should actively introduce subsidy incentive policies. For the hospital, measures should be taken to reduce costs and improve efficiency, and if necessary, they can actively seek support from the government and the park.

Given the three scenarios of the government’s management costs, $C_{gh}=\left\{10,\;40,\;60\right\}$ and $C_{gl}=\left\{0,\;20,\;40\right\}$, respectively, are shown in Figure 10 and Figure 11. On the one hand, if the government divests the management authority of the park, it will increase its management cost $C_{gh}$, and with the increase in $C_{gh}$, the evolution speed will slow down. When $C_{gh}$ increases to a specific value, the government will make the no divesture choice and let the industrial park manage the hospital. On the other hand, with the increase in $C_{gl}$, the government needs to change its strategy from no divesture to divesture so that it can obtain additional benefits from the hospital’s participation in the business environment to ensure its income. In addition, the larger $C_{gl}$, the faster such change will be.

From the perspective of cost–benefit analysis, whether the government divests the park’s management authority over the hospital will be affected by the level of the government’s management ability. That is to say, if the government is inclined to adopt the divesting policy, it should improve its management level and reduce the cost of it managing the hospital. Otherwise, merely choosing the divesture strategy to obtain additional income to cover high management costs, the government is likely to continually increase its costs instead of reducing them, which is not conducive to optimizing the business environment.

Furthermore, the trajectory of the system in the three scenarios of the government’s subsidy cost $M_{h}=\left\{0,\;40,\;60\right\}$ is shown in Figure 12. The result implies that when $M_{h}$ is small, the hospital finally chooses the nonparticipation strategy because it cannot obtain enough benefits from participating in the business environment co-creation. In this case, the industrial park chooses the strategy of low subsidy, and the government chooses no divesture. With the increase in $M_{h}$, the hospital’s strategy becomes participation, and the larger $M_{h}$, the faster the system evolves. Therefore, to encourage the hospital to choose the participation strategy, the government can try to increase its subsidy. Meanwhile, it should bear in mind that the purpose of government subsidies is primarily to inspire the hospital to be involved in business environment optimization and to deal with the interactive relationship with the industrial park. Otherwise, when the hospital chooses to participate in the co-creation, the government and the park will likely compete with each other.

Finally, the simulation result of strategy choice when the industrial park’s reputation benefits $S=\left\{0,\;40,\;60\right\}$ is shown in Figure 13. S is small, indicating the benefits the park can receive from the hospital are also very low, so it will eventually choose the low subsidy strategy. With increased S, the park is more willing to provide high subsidies to the hospital, and the greater S is, the faster the system evolution speed. It can be seen that the reputation benefits S of the industrial park have a significant influence on its strategy, and the park can make its choice by predicting S. Additionally, since the reputation benefits mainly come from the public, especially from enterprise personnel’s good evaluation and promotion of the quality of life in the park, the hospital should focus on its service satisfaction and make relevant improvements and publicity.

## 5. Conclusions 

Based on evolutionary game theory, this paper studies the substantial influence of the tripartite game decision of the government, the industrial parks, and the hospitals on the co-creation of the business environment in allusion to the present situation of the reform of the management function of industrial parks by local governments in China. This paper also analyzes how to refine and adjust the relevant policies and enhance overall regional influence in the context of high-quality development by establishing a three-party evolutionary game model. The four main conclusions are as follows: (1) the governance of the hospital in an industrial park is a representative issue in the reform of park management functions, involving the evolutionary game process of the government, the parks, and the hospitals; (2) the government chooses whether to divest the park’s administrative authority over the hospital, and the selection result is affected by the administrative costs to the government of managing the hospitals and the reputation benefits; (3) the park chooses whether to implement high subsidies for the hospitals and the selection result is influenced by the effect of hospitals’ participation in the business environment; (4) the hospital chooses whether to participate in the co-creation of the business environment, and the selection result is affected by additional benefits, subsidies, and its own participation costs. Based on the above findings, this paper puts forward corresponding recommendations.

Firstly, as hospital governance in industrial parks involves the game among multiple subjects and the selection strategies of each subject are affected by many factors, local governments should fully consider the specific situation of the factors related to local economic and social development and encourage the adoption of development measures according to the local conditions or improve relevant development conditions under the guidance of achieving high-quality development goals. It should not be a simple binary choice of whether to divest or not.

Secondly, for the government, when the cost of managing the hospital is high, the government can consider not stripping the park’s authority to manage the hospital. However, when the cost to the government of managing the hospital is not much different from that to the park of managing the hospital, especially when the extra benefits are significantly increased when the hospital participates in the co-creation of the business environment, the government can choose to divest the park’s authority to manage the hospital on the premise of avoiding unilateral interest orientation. For the industrial parks, when the hospitals participate in the co-creation of the business environment and can bring high reputation benefits, to effectively promote the high-quality development of the parks, the parks should adopt high subsidy policies. However, with the increase in subsidy cost, especially when the subsidy cost approaches the reputation benefits, the parks should attach importance to the management of the business environment co-creation of the hospital and effectively improve the co-creation effect. At the same time, the parks should appreciate communication in the interaction with the hospitals, particularly reaching an agreement with the hospitals on the strategic goal of optimizing the business environment and improving the hospitals’ recognition of the related requirements of the parks, so that the positive effect of high subsidies on the hospital’s participation can be ensured. Otherwise, it will need to consider removing the high subsidy policy. For the hospitals, when additional benefits or the parks’ subsidies increase and the cost of participation is relatively lower, the hospitals should participate in the co-creation of the business environment. Of course, from the win–win perspective, hospitals also need to consider the long-term benefits of participating in the co-creation of the business environment.

Finally, to improve the effectiveness of the co-creation of the business environment, the government should make it clear that the primary purpose of regulating and subsidizing hospitals is to encourage them to participate in the optimization of the business environment. Therefore, it is necessary to deal with the relationship between the government and the industrial parks and treat the allocation of additional benefits from the overall perspective of regional economic and social development. The industrial parks should pay attention to the quality and satisfaction of the hospital services and improve and publicize the relevant work to enhance their reputation benefits and establish a good business environment in a time-efficient manner. The hospitals should strive to improve their service level and strengthen communication with the government and the industrial parks, to reduce the cost of participating in the co-creation of the business environment and achieve win–win results.

## 6. Limitations and Prospects

The limitations of this study are mainly shown in the following two aspects: on the one hand, this paper only takes China’s industrial parks as the research object, although it can provide a reference for the management of industrial parks in other countries or regions, and the successful experience of the management of the parks in China is rapidly being imitated and copied by neighboring countries and regions [10], the different countries or regions have disparate patterns on the management of industrial parks in terms of the environment and the content of the administration, which may lead to different strategy choices of the local governments, the industrial parks, and the hospitals, and then form different game results. On the other hand, this paper only takes the comprehensive public hospitals as the representative subject. It does not consider other types of hospitals, such as hospitals for severe diseases, chronic illnesses, and clinics. In addition, regarding business environment co-creation in the parks, the social management objects include other subjects such as kindergartens and primary and secondary schools. Based on the similarity of business environment co-creation value, this paper does not study the subjects above. However, the schools and other social service providers will also show certain uniqueness. Therefore, in future research, the management rules of industrial parks in different countries or regions should be fully considered to expand the scope of the study. Meanwhile, different kinds of hospitals and schools can be introduced as game players to carry out rich and in-depth analyses.

## Figures and Tables

**Figure 1 ijerph-20-03156-f001:**
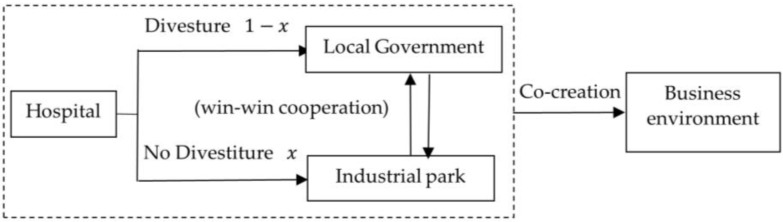
Game logic relationship in business.

**Figure 2 ijerph-20-03156-f002:**
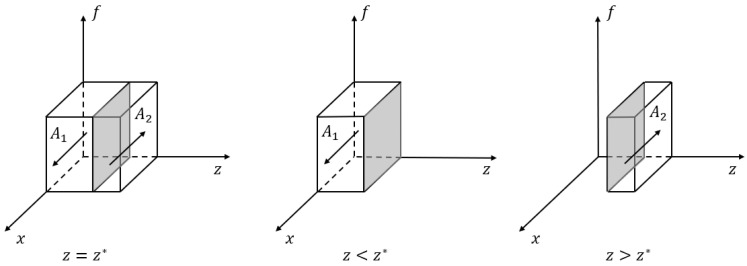
Phase diagram of strategic choices of the government.

**Figure 3 ijerph-20-03156-f003:**
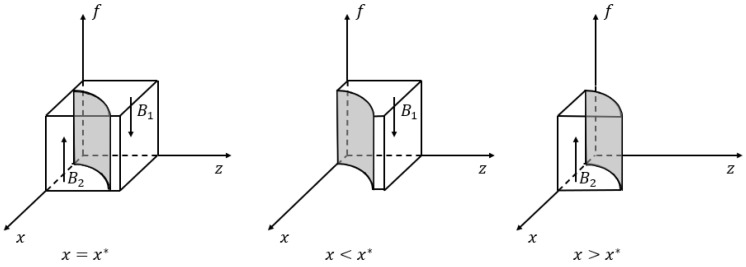
Phase diagram of strategic choices of the industrial park.

**Figure 4 ijerph-20-03156-f004:**
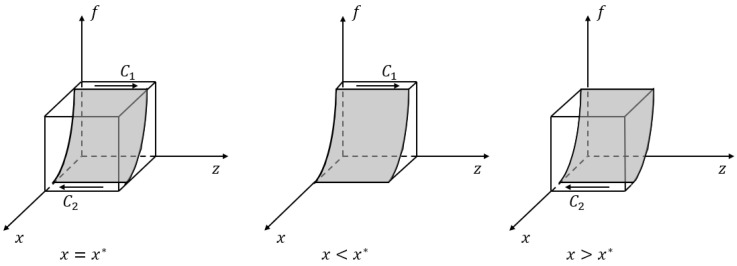
Phase diagram of strategic choices of the hospital.

**Figure 5 ijerph-20-03156-f005:**
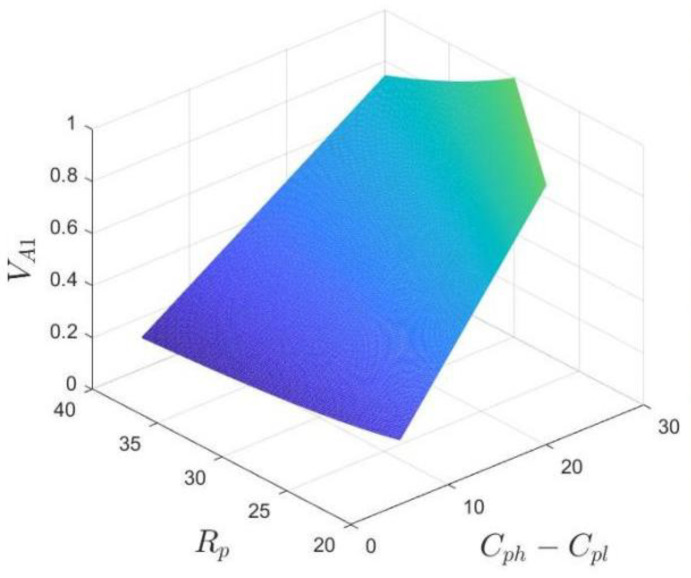
The probability of the government choosing the “ND” strategy.

**Figure 6 ijerph-20-03156-f006:**
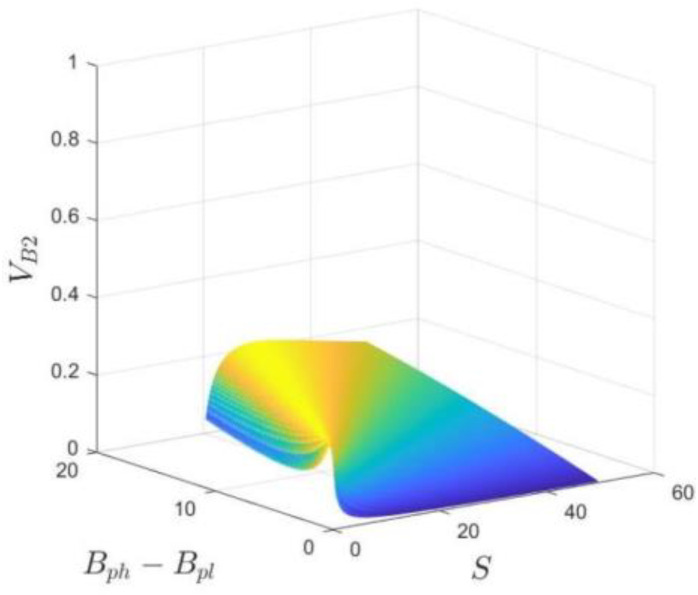
The probability of the industrial park choosing the “HS” strategy.

**Figure 7 ijerph-20-03156-f007:**
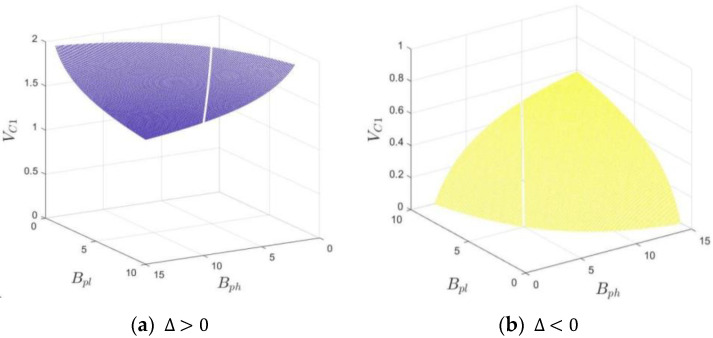
The probability of the hospital choosing the “P” strategy.

**Figure 8 ijerph-20-03156-f008:**
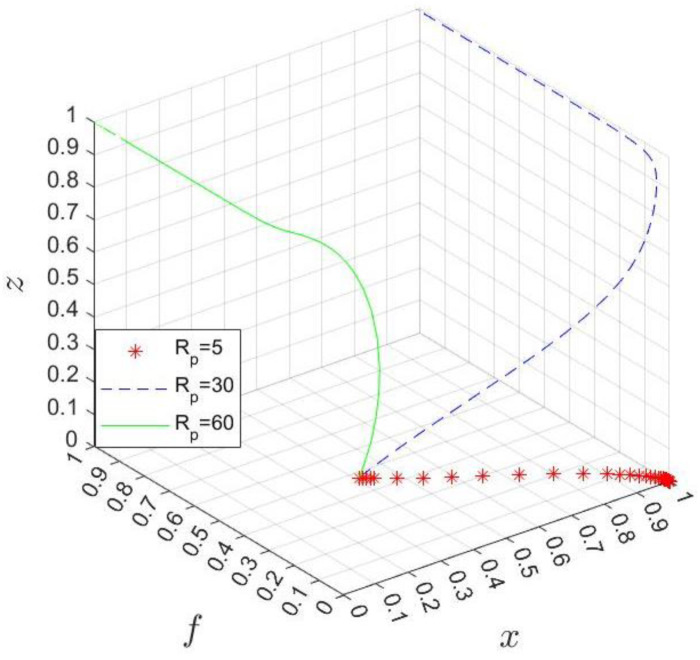
Simulation results of changing $R_{p}$.

**Figure 9 ijerph-20-03156-f009:**
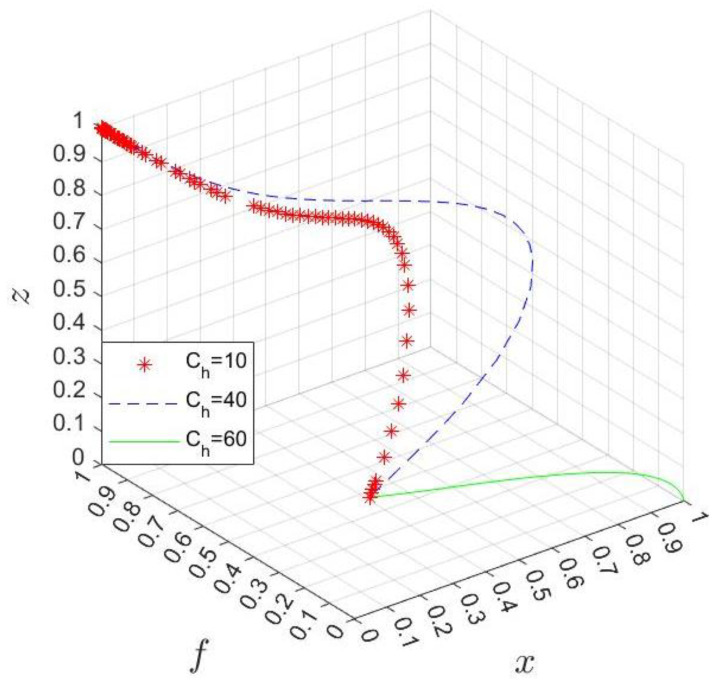
Simulation results of changing $C_{h}$.

**Figure 10 ijerph-20-03156-f010:**
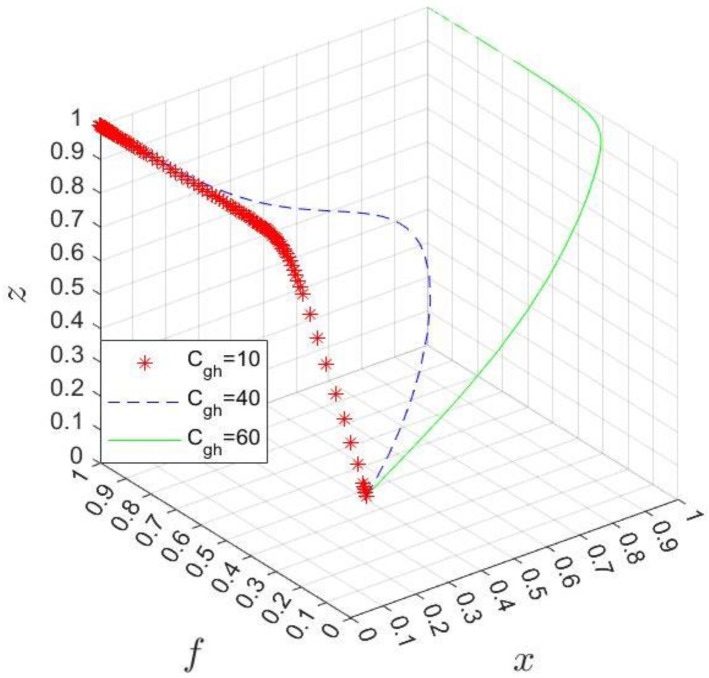
Simulation results of changing $C_{gh}$.

**Figure 11 ijerph-20-03156-f011:**
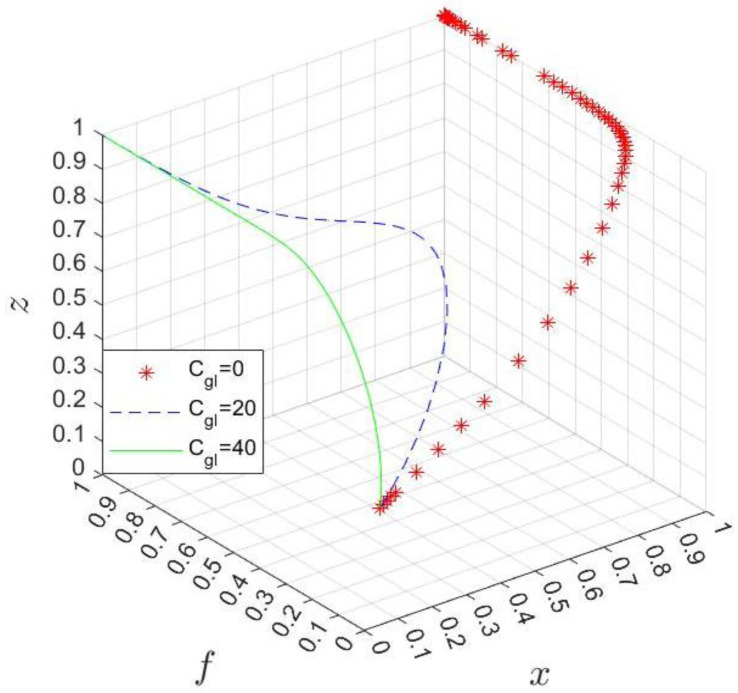
Simulation results of changing $C_{gl}$.

**Figure 12 ijerph-20-03156-f012:**
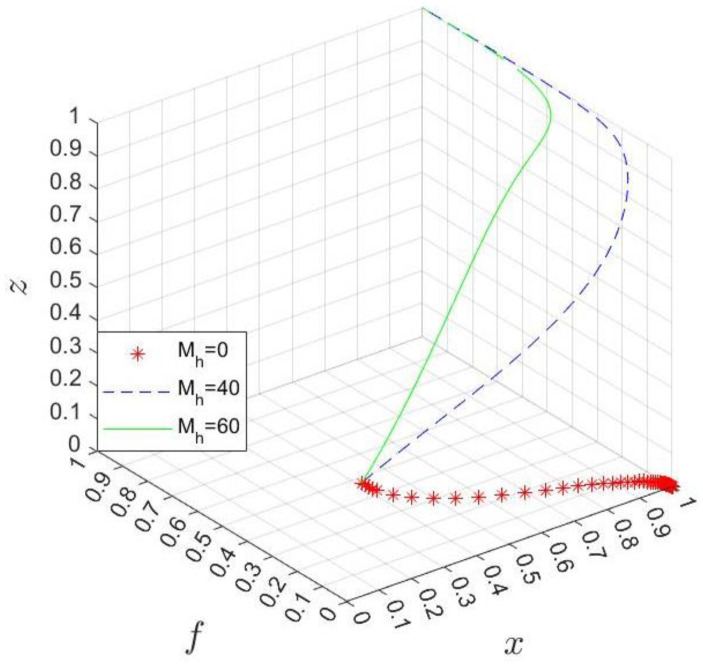
Simulation results of changing $M_{h}$.

**Figure 13 ijerph-20-03156-f013:**
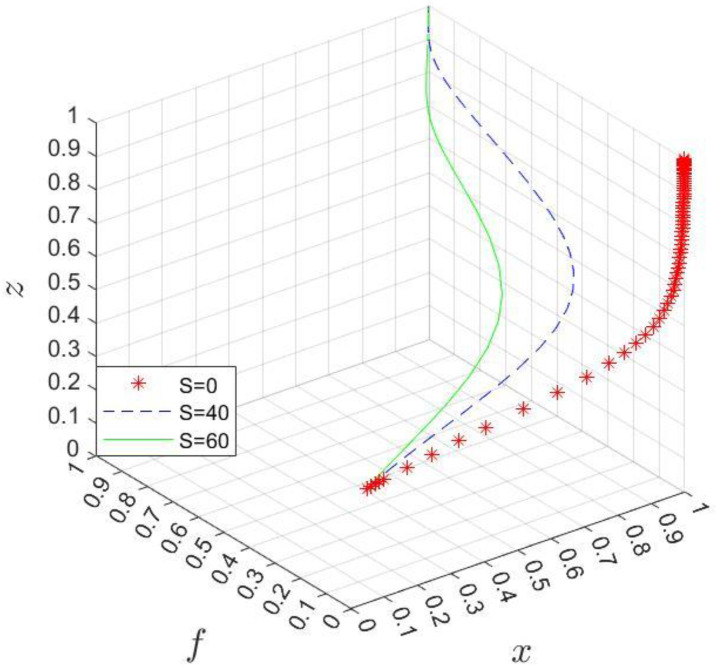
Simulation results of changing S.

**Table 1 ijerph-20-03156-t001:** Parameter settings and meanings.

Parameter	Description
$R_{p}$	the additional benefits for the hospital and its management body when participating in the co-creation of a business environment
$R_{h}$	the usual revenue of the hospital
$C_{h}$	the cost of the hospital’s participation in the business environment co-creation
$C_{gl}$	the management cost of the government to the hospital when the government chooses “ND” strategy
$C_{gh}$	the management cost of the government to the hospital when the government chooses “D” strategy
$M_{h}$	the subsidy cost of the government to the hospital that participates in business environment co-creation
$C_{pl}$	the management cost of the park to the hospital that participates in the co-creation of the business environment when the government chooses “ND” strategy
$C_{ph}$	the management cost of the park to the hospital that participates in the co-creation of the business environment when the government chooses “D” strategy
$B_{pl}$	the low subsidy cost of the industrial park to the hospital
$B_{ph}$	the high subsidy cost of the industrial park to the hospital
S	the reputation benefits of the industrial park when its strategy is “HS” and the hospital’s strategy is “P”

**Table 2 ijerph-20-03156-t002:** Evolutionary game payoff matrix of the government, industrial park, and hospital.

Game Party				Hospital	
				$P\;\left(z\right)$	$\mathrm{NP}\;\left(1-z\right)$
The government	$\mathrm{ND}\;\left(x\right)$	Industrial Park	$\mathrm{HS}\;\left(f\right)$	$(-C_{gl}-M_{h},S+R_{p}-C_{pl}-B_{ph},\;R_{h}+R_{p}+M_{h}+B_{ph}-C_{h}$ *)*	$\left(-C_{gl},-B_{ph}-C_{ph},R_{h}+B_{ph}\right)$
			$\mathrm{LS}\;\left(1-f\right)$	$\left(-C_{gl}-M_{h},\;R_{p}-B_{pl}-C_{pl},\;R_{h}+R_{p}+M_{h}+B_{pl}-C_{h}\right)$	$\left(-C_{gl},-B_{pl}-C_{ph},R_{h}+B_{pl}\right)$
	$D\;\left(1-x\right)$	Industrial Park	$\mathrm{HS}\;\left(f\right)$	$\left(R_{p}-C_{gh}-M_{h},S-B_{ph},R_{h}+R_{p}+M_{h}+B_{ph}-C_{h}\right)$	$\left(-C_{gh},\;0,\;R_{h}\right)$
			$\mathrm{LS}\;\left(1-f\right)$	$\left(R_{p}-C_{gh}-M_{h},-B_{pl},\;R_{h}+R_{p}+M_{h}+B_{pl}-C_{h}\right)$	$\left(-C_{gh},\;0,\;R_{h}\right)$

**Table 3 ijerph-20-03156-t003:** Stability analysis of evolutionary strategic portfolios.

Equilibrium	Eigenvalue *λ*_1,_ *λ*_2,_ *λ*_3_	Sign	Stability
$\;\;\;\left(0,\;0,\;0\right)$	$C_{gh}-C_{gl}$ 0 $,\;R_{p}+M_{h}-C_{h}-B_{pl}$	(+, 0, u)	Unstable
$\;\;\;\left(0,\;0,\;1\right)$	$-R_{p}+C_{gh}-C_{gl}$ $,\;S-\left(B_{ph}-B_{pl}\right)$ $,-R_{p}-M_{h}+C_{h}+B_{pl}$	(u, +, u)	Unstable
$\;\;\;\left(0,\;1,\;0\right)$	$C_{gh}-C_{gl}$ 0 $,\;R_{p}+M_{h}-C_{h}+B_{ph}$	(+, 0, u)	Unstable
$\;\;\;\left(0,\;1,\;1\right)$	$-R_{p}+C_{gh}-C_{gl}$ $,-S+\left(B_{ph}-B_{pl}\right)$ $,-R_{p}-M_{h}+C_{h}-B_{ph}$	(u, −, u)	ESS under Condition 1
$\;\;\;\left(1,\;0,\;0\right)$	$-C_{gh}+C_{gl}$ $,-B_{ph}+B_{pl}$ $,\;R_{p}+M_{h}-C_{h}$	(−, −, u)	ESS under Condition 2
$\;\;\;\left(1,\;0,\;1\right)$	$R_{p}-C_{gh}+C_{gl}$ $,\;S-\left(B_{ph}-B_{pl}\right)$ $,-R_{p}-M_{h}+C_{h}$	(u, +, u)	Unstable
$\;\;\;\left(1,\;1,\;0\right)$	$-C_{gh}+C_{gl}$ $,\;B_{ph}-B_{pl}$ $,\;R_{p}+M_{h}-C_{h}$	(−, +, u)	Unstable
$\;\;\;\left(1,\;1,\;1\right)$	$R_{p}-C_{gh}+C_{gl}$ $,-S+\left(B_{ph}-B_{pl}\right)$ $,-R_{p}-M_{h}+C_{h}$	(u, −, u)	ESS under Condition 3

Conditions: 1. $C_{gh}-C_{gl}<R_{p}$, $C_{h}<R_{p}+M_{h}+B_{ph}$; 2. $R_{p}+M_{h}<C_{h}$; 3. $C_{gh}-C_{gl}>R_{p}$, $C_{h}<R_{p}+M_{h}$.

**Table 4 ijerph-20-03156-t004:** Initial values of simulation parameters.

Parameters	$R_{p}$	$C_{h}$	$C_{gh}$	$C_{gl}$	$M_{h}$	$B_{ph}$	$B_{pl}$	S
Values	40	35	40	10	10	10	5	20

## Data Availability

Not applicable.

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
