# Peer review of "Evolution Game Analysis of Hospital Governance Strategy in Industrial Parks in China"

_ijerph, 2023, doi:10.3390/ijerph20043156_

Round 1

Reviewer 1 Report

Even though there are many different types of hospitals, such as acute hospitals, chronic, and  clinics, we could not find the definition.

Author Response

Dear Reviewer:

Many thanks for your valuable review comments. As you suggested, we have revised the paper and written the modification response. Please kindly find the attachment for the description of the modification.

Best wishes,

The authors

Reviewer 2 Report

While I acknowledge that the manuscript would be of interest to some (small) specific group of readers, I must conclude that it is by far not suitable for a scientific journal.

The problems start already with the title.  It is very long and complicated. Especially the concept of "Park Management Function Reform" is hard to understand at first sight, and is neither clearly discussed in the manuscript.

The abstract is still worse.  It is not giving an adequate image of the manuscript, and is hard to follow.  A structure abstract say for example in APA style is needed:  How to write an APA abstract

  1. What is the problem? Outline the objective, research questions, and/or hypotheses.
  2. What has been done? Explain your research methods.
  3. What did you discover? Summarize the key findings and conclusions.
  4. What do the findings mean? Summarize the discussion and recommendations.

The other main shortcomings of the article are the following:

1 the research gap is not described at all.  

2 The research problem is not formulated at all in an condensed way.  There is no short research question described.

3 The article contains no discussion on theory, say for example on "Hospital Governance Strategy" or any similar central concept of the work

4 The method "Evolution Game Analysis" is not described at all, and the reasons for using it are not given.  The long and detailed mathematical work all through is not at all displacing this need, the mathematical work in the paper rather blinds than illuminates

5 it is hard to see how the conclusions are drawn from the (speculative game) analysis (based on author views only).  The conclusions drawn do not feed back to theory (as there was not any) or neither to practice.  It remains too totally undiscussed how one can extend the conclusions to outside China. 

6 There is not at all indices of the normal discussions on limitations of the work and further research avenues.

6 There is no discussion why the hospital is taken to the core of analysis.

The article suffers from bad English everywhere, probably because of bad direct translation to Chinese to English.

Author Response

Dear reviewer:

Many thanks for your valuable review comments. As you suggested, we have revised the paper and written the modification response. Please kindly find the attachment for the description of the modification.

Best wishes,

The authors

Reviewer 3 Report

The topic is relevant. The research is logical, the conclusions are clear, logical and even predictable. There are 4 self-citations of the author, but they all correspond to the topic. Decipher the abbreviation ‘GDP’ in line 32

Author Response

(The authors gave the same response as above.)

Author Response

(The authors gave the same response as above.)

Round 2

Reviewer 1 Report

All points raised by the peer reviewers were revised, corrected, and added.